# Research on Open Practice Teaching of Off-Campus Art Appreciation Based on ICT

**Baoqing Song** [1,2], **Bingyang He** [2], **Zehua Wang** [2] , **Ruichong Lin** [2] , **Jinrui Yang** [3], **Runxian Zhou** [4] **and Yunji Cai** [5,*]

1. Art Education Center, Beijing Institute of Technology Zhuhai, Zhuhai 519085, China; song_bq@bitzh.edu.cn
2. Graduate School, Beijing Normal University-Hong Kong Baptist University United International College, Zhuhai 519087, China; r130233083@mail.uic.edu.cn (B.H.); r130233077@mail.uic.edu.cn (Z.W.); r130233046@mail.uic.edu.cn (R.L.)
3. School of Information Technology, Beijing Institute of Technology Zhuhai, Zhuhai 519085, China; yjr0729@126.com
4. School of Design and Art, Beijing Institute of Technology Zhuhai, Zhuhai 519085, China; chantrop@163.com
5. Student Affairs Office, Beijing Institute of Technology Zhuhai, Zhuhai 519085, China
* Correspondence: cai_yj@bitzh.edu.cn

**Abstract:** Art appreciation is an effective way to promote artistic literacy and is also an important component of aesthetic education in school. With the help of information and communication technology, the authors organized open practice teaching for students to learn art appreciation outside school. During the COVID-19 epidemic, local art appreciation education could not be carried out in the city where the authors' school is located. With the support of mobile positioning technology and information platforms, students were able to carry out 32 art appreciation activities in their hometowns during this period. Through the mobile positioning information submitted by students, feedback questionnaires, after-view data, and other data, learning achievements were identified. A correlation analysis of the data submitted by the students on the information platform confirmed that satisfaction with the art appreciation activity directly affected their interest in art. The correlation reached 0.78. Satisfaction was strongly correlated with psychological expectations (0.67) and art information obtained in the early stage (0.61). The authors propose that using information and communication technology to carry out art appreciation education outside the school is the way to promote the sustainable development of aesthetic education in school.

**Keywords:** ICT; OEP; aesthetic education; SRL; university education

## 1. Introduction

Since the COVID-19 outbreak in 2020, some changes have taken place in the field of education, and open education practices (OEP) have received more attention. Schools in countries that have been severely affected by the epidemic have adopted forms of distance teaching and use online teaching to keep students from being suspended from school [1–4]. Lecture-type teaching can be carried out using self-study in the form of MOOCs [5] or Zoom and other remote tools, and indoor practical teaching also yields encouraging results, according to a French study [6,7]. However, for practical teaching content with strong outdoor mobility, such as outdoor painting, art appreciation, etc., the support of information and communication technology is indispensable [8], including mobile positioning, technology, information platform application [9], and so on. This research group is responsible for the management of art education in school and focuses on exploring the teaching mode of practical art appreciation. Practical art appreciation plays a significant role in students' understanding of art [10] and the development of active learning habits [11]. This work was aimed at teaching out-of-school art appreciation supported by an information technology platform. Since there are also some practical hours

in other courses, this research has greater application value during the pandemic. This research has the following two main purposes:

- Can students appreciate art outside of school through self-directed learning (SRL) and enhance their artistic interest?
- Based on ICT support, can students achieve good learning outcomes and teaching assessment requirements?

This study attempts to provide some answers to these questions. From January to February 2022, 32 students conducted art appreciation activities in several different provinces in China, which included painting exhibitions, cultural relics visits, etc. Since art appreciation activities are scattered, teachers will not manage students one-to-one. Through the Feishu app, they will record the viewing time, viewing location, viewing content, etc. of students and collect feedback on students' learning through questionnaires to obtain an assessment of students' learning quality through data analysis.

The article is divided into five sections. Section 2 introduces the research results of autonomous learning (SRL) in distance education, art appreciation in aesthetic education, the background of China's aesthetic education policy, and the previous research results of our research group. Section 3 introduces the research methods and materials. Section 4 describes the experimental results. Section 5 discusses the experimental results and other research results.

## 2. Related Backgrounds

### 2.1. SRL in the Distance Education

SRL refers to the active and active behavior of individuals in learning, which is one of the important topics in the field of pedagogy [12]. Zimmerman proposed a widely accepted fundamental theory of self-generated thoughts, feelings, and behaviors that are systematically guided by personal goals [13]. During the COVID-19 pandemic, countries around the world have adopted campus closure measures, and students study remotely, which means the research of SRL theory in distance education has received further attention. Under the situation of distance education, students must accept SRL [14], which requires the guidance and help of teachers. In addition, utilizing the large amount of educational data generated in the online learning environment can help to understand the learning needs of students [15] and adjust teaching strategies appropriately.

### 2.2. Aesthetic Education and Art Appreciation

Aesthetic education covers a wide range, among which art appreciation is the most direct way to promote the aesthetic level. In the process of appreciation–thinking–evaluation–learning, people's aesthetic level is constantly improved, and the "Aesthetic Quotient" (AQ) accumulated on this basis is also improved [16]. The research from Funch showed that there are five different types of art appreciation (aesthetic pleasure, emotional appreciation, cognitive appreciation, aesthetic charm, and high consciousness), and the aesthetic experience gained from art appreciation is the most personalized participation a person may have in a work of art [17]. The description of intense experience in works of art shows that an extraordinary sensory appearance is followed by an emotional quality beyond ordinary consciousness, and this connection between special sensory forms and emotional response not only indicates temporary enjoyment but also indicates emotional reorganization [18]. Spiritual sublimation through artistic appreciation is one of the main purposes of aesthetic education.

### 2.3. Policy Background

In October 2020, the Chinese government released the latest aesthetic education policy, which imposes strict requirements on colleges and universities. Each student must complete two art credits before graduation. In addition to the requirements for students' learning, the policy also imposes rigid requirements for implementation by schools. School administrators who fail to implement the policy will be held accountable, and schools are

not allowed to apply for double first-class [19]. Since 2005, China's provincial education departments, the Ministry of Education, the State Council, and the Central Committee of the Communist Party have successively issued policies on art education, and their dissemination has gradually increased, indicating that China's emphasis on students' aesthetic education has been increasing.

However, due to the long-term neglect of aesthetic education, there is a shortage of art teachers at colleges and universities. According to the policy requirements, a university with a population of 20,000 should be equipped with no fewer than 35 art teachers, including music, dance, art, calligraphy, film and television, drama, and other major art categories. This is difficult for schools with a weak aesthetic education foundation. For this reason, some schools hire social artists to teach part of the courses [20], and some schools supplement the number of courses in the form of MOOCs [21–23], but these two supplementary mechanisms have certain deficiencies, such as insufficient teaching skills of artists, insufficient content of MOOCs, etc. [24]. In addition to these methods, the university where the authors are located is exploring a new path of practical aesthetic education, with art appreciation as the main content.

### 2.4. Preliminary Research Results

From March to October 2021, the research team carried out a series of explorations on school teaching and invited nonacademic professional art groups to come to schools to offer practical art appreciation courses. These courses were welcomed by students, and participating students were awarded credit for taking them. In a demand survey based on the Kano model [25–27], it was found that students (N = 976) had the highest interest in art appreciation courses with professional teams as the main instructors [28]. It was also found in the survey that students had higher expectations for professional venues such as theaters and concert halls compared to customized art appreciation courses on campus.

In December 2021, the research team randomly selected 51 students from among those who participated in the on-campus art appreciation course and organized a concert at a local professional theater for them to enjoy. The joy of art appreciation at off-campus professional venues is even higher. Appreciating professional performances in such venues makes students' perceptions and experience of art more complete [29]. It was found that four out of five of the students tended to combine on-campus art appreciation courses with art appreciation at professional off-campus venues. At the same time, more than half of the students preferred to use mobile phone positioning for attendance (Figure 1).

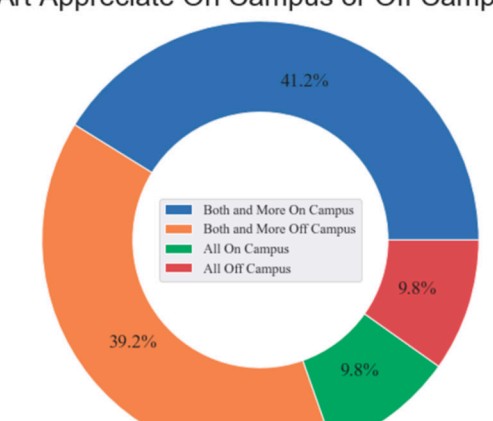
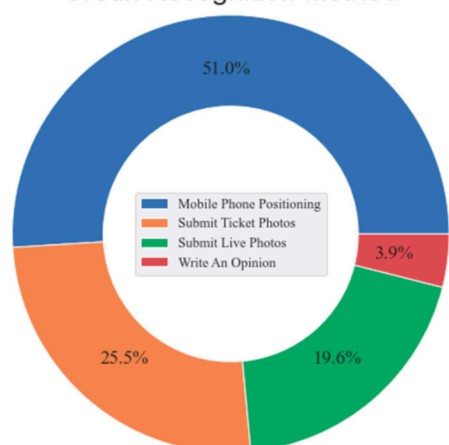

**Figure 1.** Preliminary research results.

### 2.5. The Purpose of This Research

In China's aesthetic education policy, in addition to schools being encouraged to supplement aesthetic education with the two methods described above, colleges and uni-

versities are also encouraged to cooperate with social, cultural, and art venues, and students are encouraged to go to concert halls, art galleries, and museums to experience the arts in person. In this process, students have ownership and control over their learning in the form of SRL; they can choose art appreciation content according to their own plans or randomly [30] and can achieve real-time feedback and time management [31]. Zimmerman's cyclical model [32] is formed by the process of students selecting viewing content, developing a viewing plan, viewing the content, and conducting a post-view review, during which they improve their artistic literacy through their own actions, and guide, standardize, and evaluate their own learning [33]. Based on this, the study assumed that curricular art appreciation outside the university can promote students' interest in art.

The other purpose of this experiment was to verify the effectiveness of teaching on an information platform for autonomous learning outside school. While being on campus can ensure the normal teaching order through teachers' on-site sign-in, off-campus art appreciation is limited by venue facilities, where it is difficult to achieve on-site sign-in. With the help of mobile communication positioning technology [34] and the application of mobile terminal information platform recording [35–37], which is an optimal choice to ensure the teaching order, the research team conducted an independent art appreciation experiment with some students in January 2022. It was assumed that, based on the support of information and communication technology, students could obtain good learning results with scattered and independent art appreciation activities and meet the requirements of teaching evaluation.

Due to the sudden outbreak of COVID-19 in Zhuhai, Guangdong, China, where the school is located, all indoor public art venues were temporarily closed, interrupting the original plan. The research team discussed that the factors of the epidemic may have a special role in the evaluation of the study results [38]. Due to China's precise prevention and control measures [39], the opening of public indoor places is only controlled in medium- and high-risk areas, while exhibition halls and theaters in most low-risk cities can still open normally. The research team decided to take advantage of the Chinese New Year holiday to organize the experiment remotely in their hometowns. After students leave the city where the school is located, the nature of their art appreciation learning is more social, random, and accidental, making the experimental results obtained more representative [40].

## 3. Materials and Methods

### 3.1. Experimental Procedure

Off-campus art appreciation education adopts an open practice teaching mode. According to their preferences, students can independently choose the content of the performance or exhibition, the venue, the study time, and their companion. To ensure the authenticity of the learning content, the length of learning, and the learning effect, the research team relied on the information platform to set up the learning check-in and feedback process, as shown in Figure 2. For example, first, student A looks at the recent exhibition information of "Minying Art Museum" in his area, selects "SunYue's Personal Art Exhibition" that he is interested in, check in with Feishu at the scene, and locates his location in this art museum. After confirming the sign-in, choose the sign-in questionnaire, fill in the information of this exhibition, upload the photos of the scene, and then enjoy the exhibition. After the appreciation, check out through Feishu, and fill out the sign-out questionnaire, including the evaluation and feelings of this exhibition. The research team used the location information and questionnaire data derived from the Feishu background to analyze the data matching, judged the viewing feeling manually, and finally recognized credits.

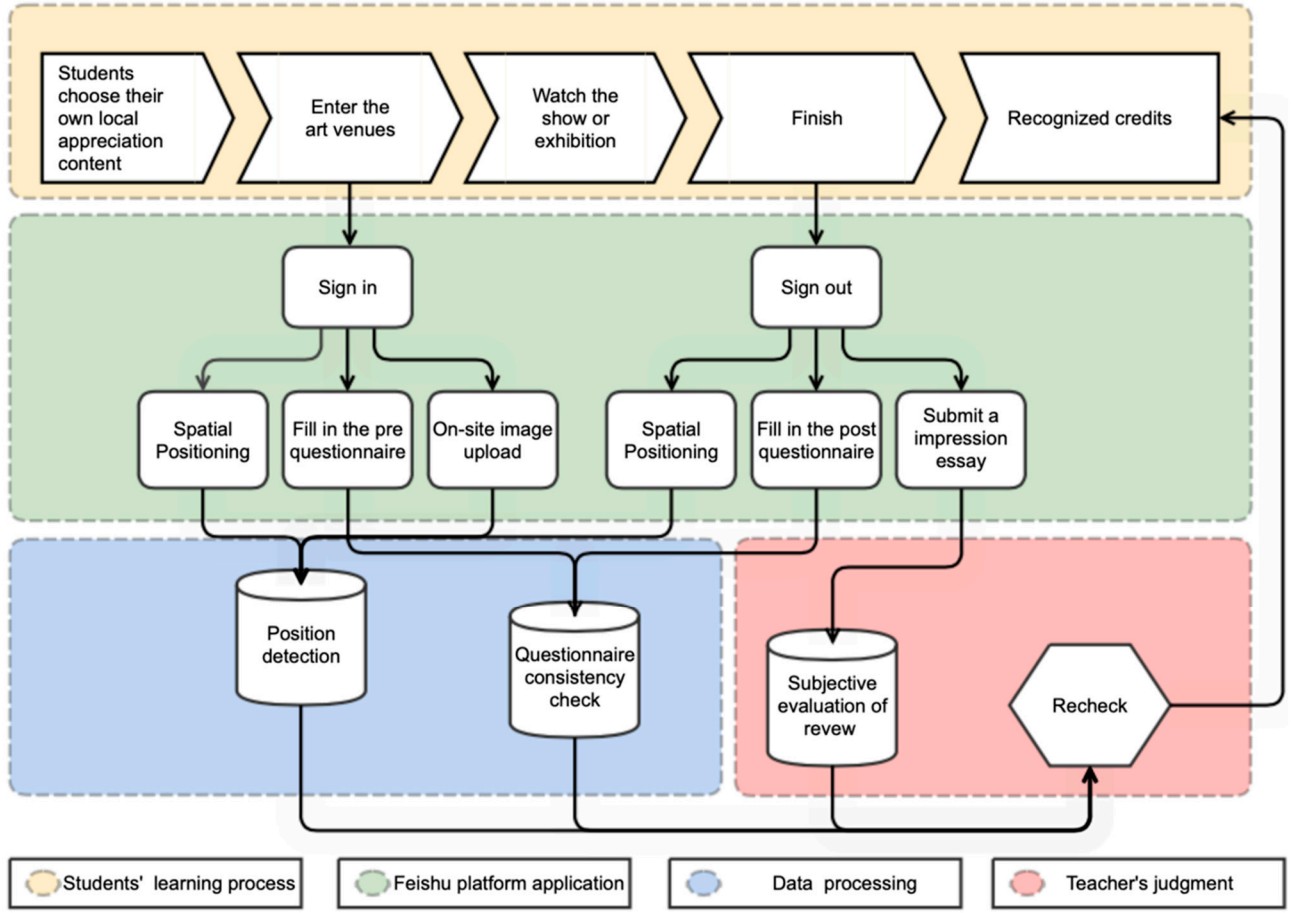

**Figure 2.** Experimental procedure.

### *3.2. Mobile Positioning and Information Platform*

Based on GPS and China's Beidou satellite navigation system [41], mobile phones are used as positioning terminals with high positioning accuracy, and most theaters, concert halls, and exhibition halls in most cities in China offer free Wi-Fi. Fingerprint recognition technology complements indoor positioning [42–44].

This experiment used the Feishu app developed by ByteDance as the main information platform. This tool has functions such as location check-in, information input, and picture upload, which can collect the information records of students' art appreciation activities. According to the trial feedback, the group administrator summarized the art appreciation activity information for all participating students through the app, which is convenient for verification, statistics, and analysis. According to the experimental needs, the research team developed two sets of questionnaires for check-in before the activity (Figure 3) and check-out after viewing (Figure 4), which could meet the basic needs of the teaching evaluation.

### *3.3. Experimenter*

In the main part of this experiment, 32 students signed up to participate. In consideration of the ethics of social science research, members of the research team will introduce the process and purpose of the research project to the participating students in detail and inform the students of their rights and answer their questions. Students can voluntarily withdraw from the experiment at any time without any penalty. In line with the best experimental model proposed by Dr. Petousi and Dr. Sifaki [45], during the holiday from January to February 2022, they chose performances and exhibitions of interest in their hometowns to watch. Participating students received 0.2 art credits for each art appreciation project.

They were required to check in on the Feishu platform by mobile location before the activity and fill out a questionnaire containing the category of the art project, the subject name, the sponsoring team, and their psychological expectations. Questionnaire items included meeting expectations, after-view satisfaction, etc. To ensure normal development of the experiment, according to Bravo's experience [46], the research team conducted detailed training with the participating students, including the use of the information platform and descriptions of the questionnaire options. The research team also designated a special person to be responsible for the online Q&A during the experiment and regular maintenance of the data on the information platform.

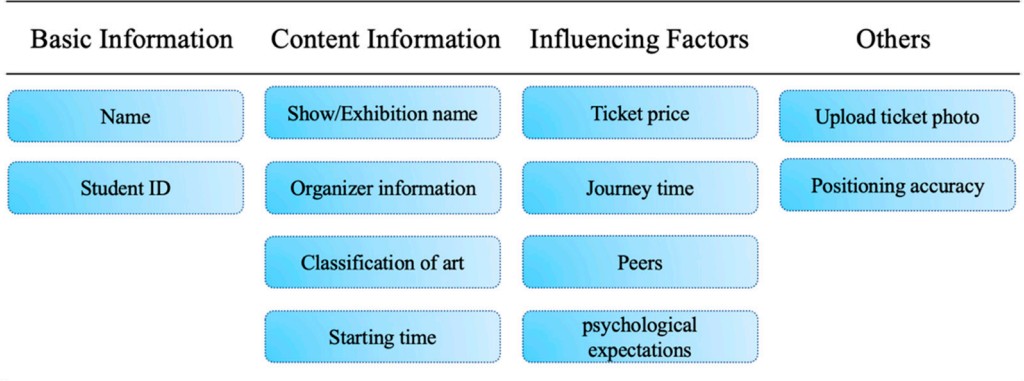

**Figure 3.** Check-in questionnaire content.

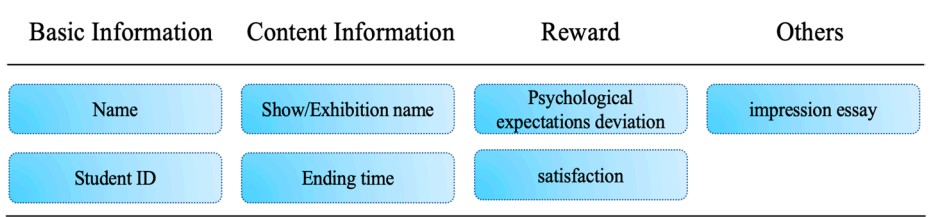

**Figure 4.** Check-out questionnaire content.

### 3.4. Data Processing Method

Export students' two positioning information data and two questionnaires information from the Feishu management background. Using Pandas combines these two sets of data. Consistency checks are performed on student numbers, positioning data, and appreciation item names to ensure the accuracy of metadata.

Taking improved interest, as reported on the check-out questionnaire, as the main indicator, using Pearson's correlation and significance analysis methods, the options strongly correlating with it were analyzed, and factors that affect students' off-campus art appreciation were obtained, and the key path for further improvement of this learning model was found.

Pearson correlation analysis is widely used in part of the data involved, especially for the correlation of certain characteristics between people. In the design of this paper, we also analyzed the variables based on users, especially those who are more closely related and using Pearson correlation analysis is a more reasonable choice; considering the small sample size of this experiment, we also passed the data through the Pearson test, which also ensured the independence between variables. In addition to the Pearson correlation analysis, there are also Spearman correlation coefficients and Kendall correlation coefficients, which are measures of order correlation and correlation with ordered categories, respectively, and are not applicable in the analysis of this paper, so these two means of correlation statistical analysis were not used.

## 4. Results

### 4.1. Experiment Results

According to the statistics of the information platform, all students who participated in the experiment carried out 32 art appreciation activities in their hometowns, which were distributed in eight provinces in China, as shown in Figure 5. Among the students, 93.75% were able to sign in and out during the check-in and check-out process. The type of activity that students chose most frequently was cultural relics appreciation, accounting for 56.25% of the total. Students were generally satisfied with the art appreciation activities they participated in, and 53.12% of them reported they were very satisfied with the activities. In terms of the level of art appreciation content, the majority was at the provincial level, accounting for 59.38%. The detailed activity statistics are shown in Figure 6.

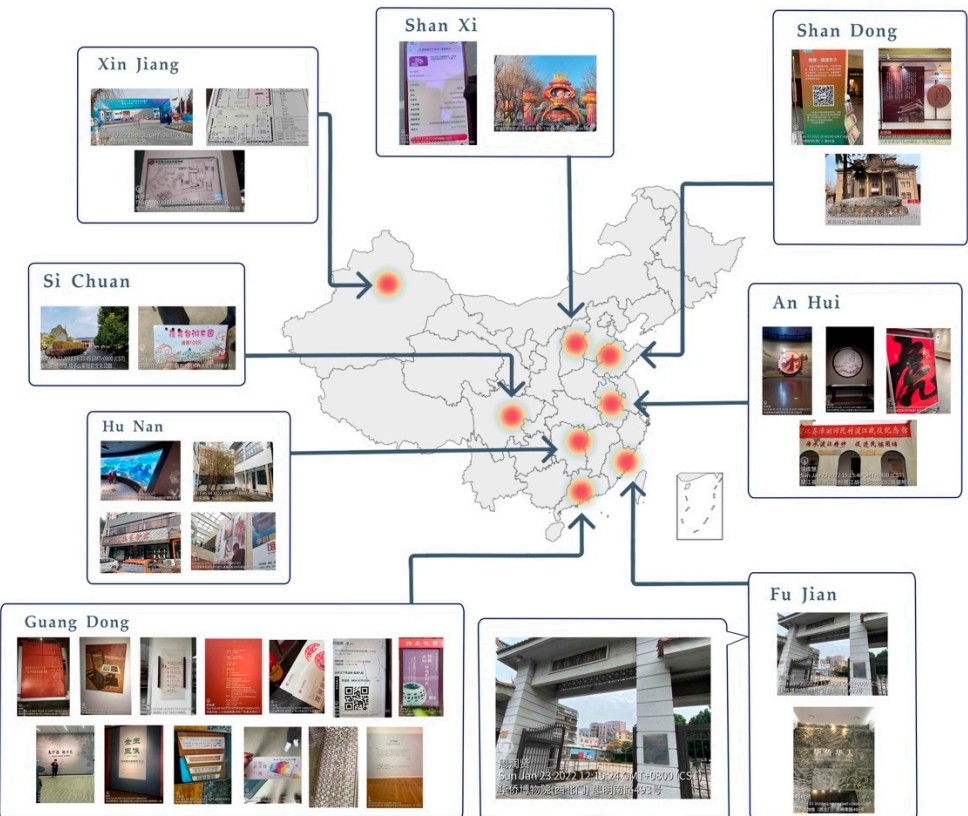

**Figure 5.** Map of activity area distribution.

### 4.2. Data Analysis Results

First, data matching of the check-in and check-out questionnaires was performed to obtain the complete data. Since the sample size was 32, the lower limit of the Pearson's test was reached, so the Shapiro–Wilk test, which is suitable for small samples, was used to verify the data. According to the data calculations, $p > 0.05$, indicating that the data had a normal distribution to meet the requirements for further analysis. Using the Pearson correlation test, it was found that the highest correlation with increased interest was satisfaction, and the two factors that were highly correlated with satisfaction were psychological expectations and art appreciation information, as shown in Figure 7.

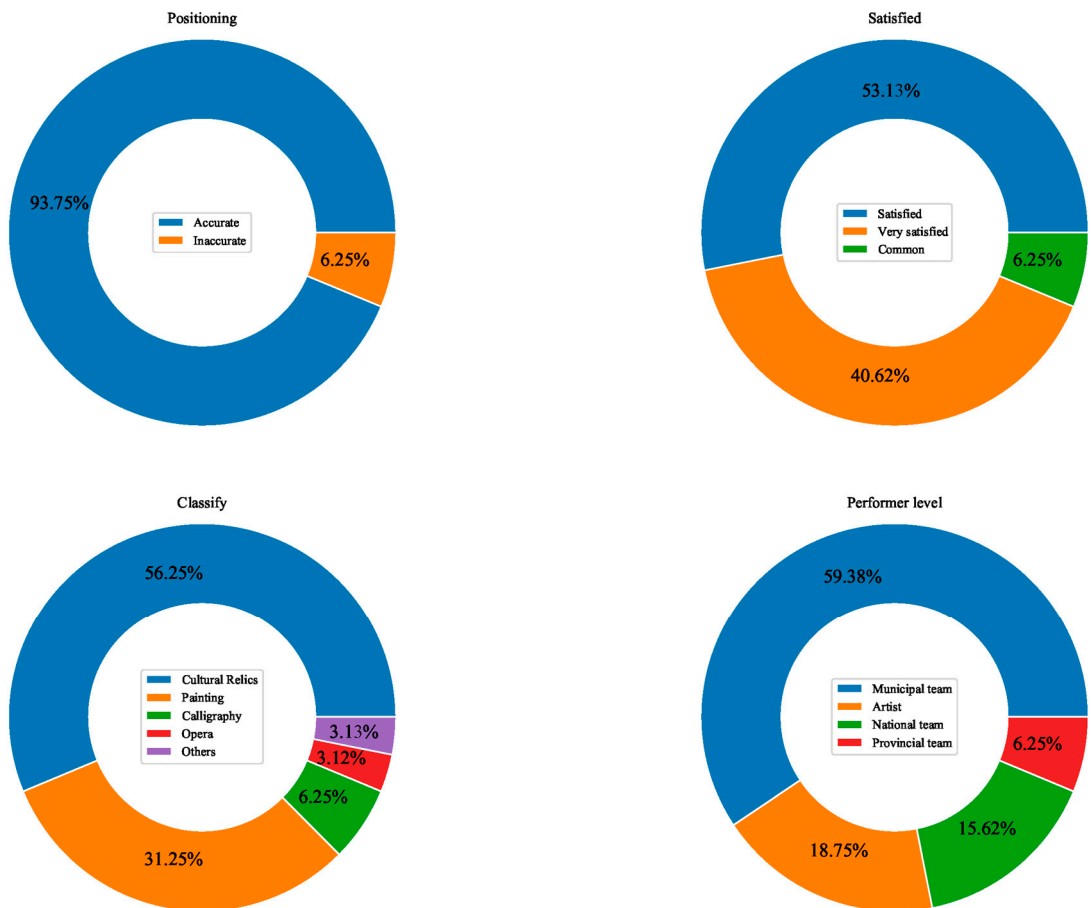

**Figure 6.** Activity information statistics.

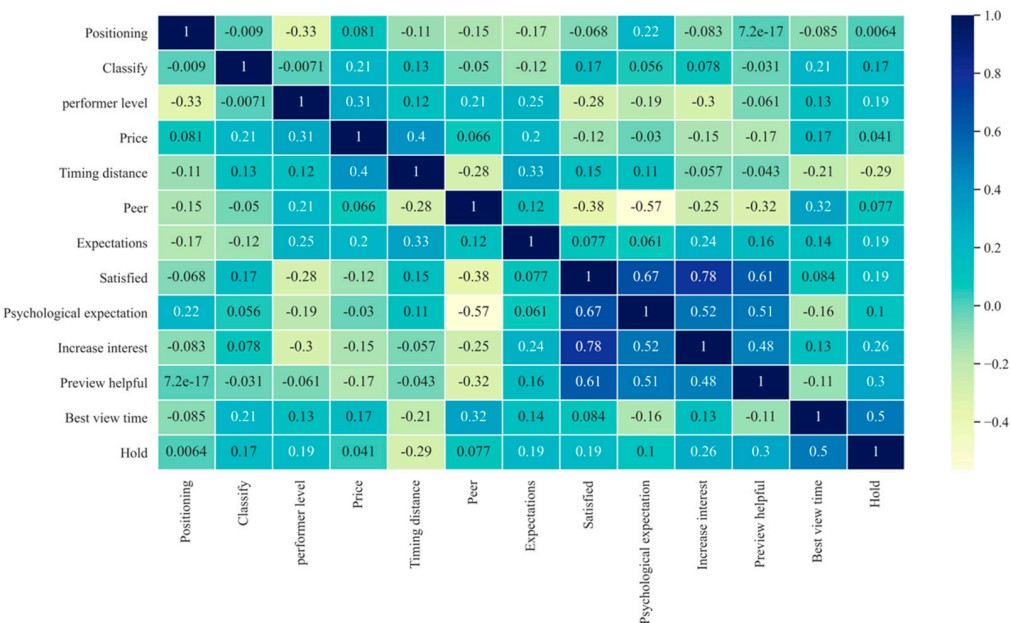

**Figure 7.** Correlation analysis.

Using Pearson's significance test, it is found that a student's interest in art appreciation had the strongest correlation with satisfaction, and psychological expectations and art appreciation information also had a strong correlation with satisfaction, which is consistent with the correlation analysis results (Table 1). A visual presentation of interest–satisfaction,

satisfaction–expectation, and satisfaction–art information preview for a more intuitive understanding of the degree of correlation is shown in Figures 6–8. According to the results of the Pearson analysis, we visualized the different variables linearly with the satisfaction we eventually want to study, where each circle indicated the more concentrated area of the data, i.e., the distribution of the data referred to by the *x*-axis and *y*-axis; for example when x has two peaks and y also has two peaks, there will be four central circles in the graph, as shown in Figures 8–10. It can be seen that Increase interest distribution center is basically linearly correlated with student satisfied centroid in Figure 8, while Psychological expectation and Preview helpful can also be seen to be positively correlated with student satisfied center distribution in Figures 9 and 10, but obviously, the linear association is not as strong as the increase interest. This is also understandable, because generally speaking, if a user is more satisfied, he or she will naturally be more interested in the item, but high user satisfaction does not necessarily mean that the item is helpful in a practical sense but may only have a psychological transient effect.

**Table 1.** Pearson's significance test.

|  | 1 | 2 | 3 | 4 | 5 | 6 |
|---|---|---|---|---|---|---|
| **Item A** | Increase interest | Psychological expectations | Preview helpful | Psychological expectations | Increase interest | Preview helpful |
| **Item B** | Satisfied | Satisfied | Satisfied | Peer | Psychological expectations | Psychological expectations |
| $p$ | $1.89404582104268 \times 10^{-7}$ | $3.14581603588882 \times 10^{-5}$ | $1.87530180025815 \times 10^{-4}$ | $7.43520014047949 \times 10^{-4}$ | $2.33872682150937 \times 10^{-3}$ | $2.90517692166116 \times 10^{-3}$ |

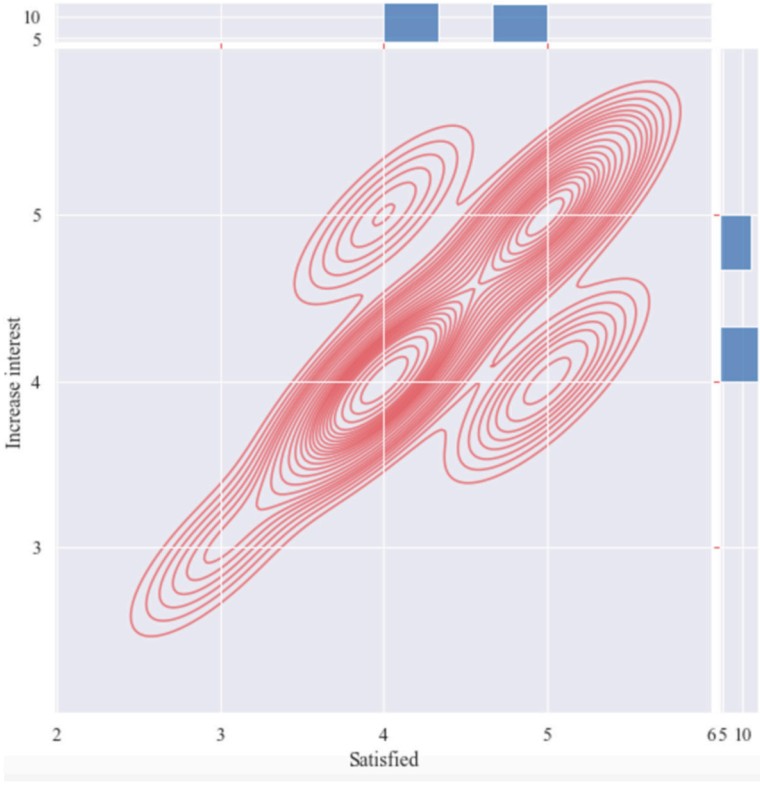

**Figure 8.** Interest–satisfaction.

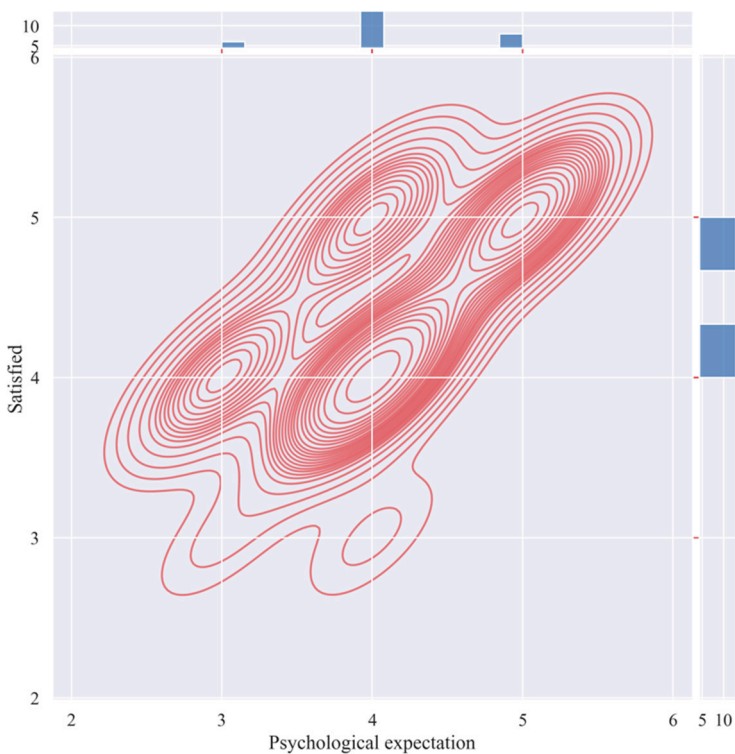

**Figure 9.** Satisfaction–expectation.

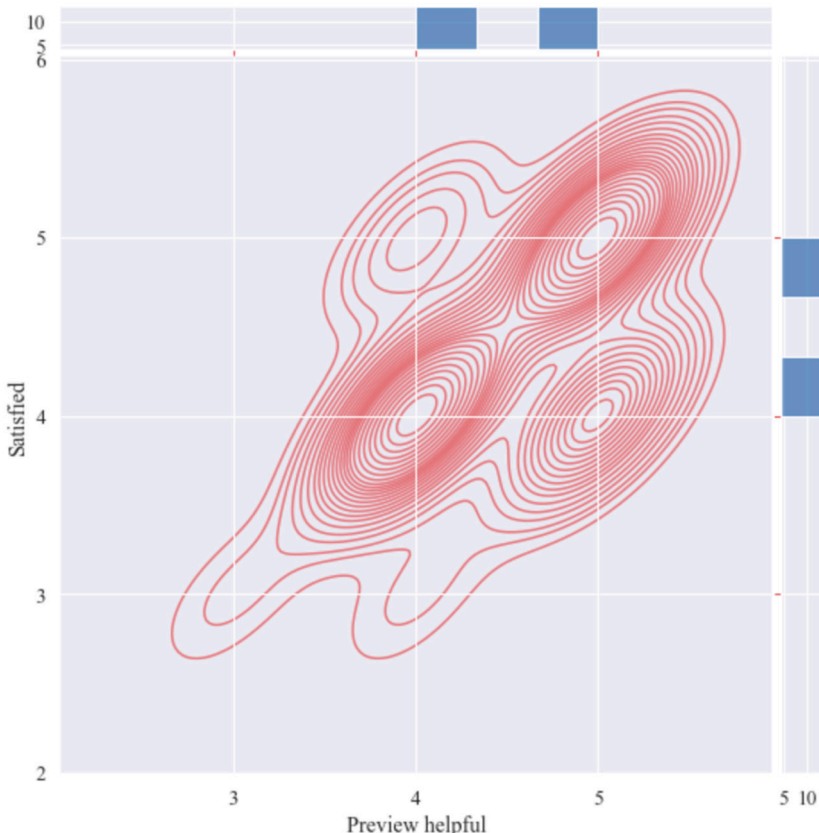

**Figure 10.** Satisfaction–art information preview.

## 5. Discussion

From the analysis of the results, an improvement in students' interest in art appreciation came from satisfaction with the viewing content, and satisfaction was closely related to psychological expectations and the introduction of the art appreciation content. This is consistent with the research of Abe and Sova, with similar conclusions [47,48]. In interviews with individual students, it was found that when they carried out art appreciation activities, they could only obtain information on the activity on the spot, and the ability to expand the relevant art knowledge was limited, which limited critical thinking to a certain extent. Under normal school conditions, if the city where the university is located carries out this kind of off-campus art appreciation study, it may have a better learning effect. Through cooperation between schools and government cultural departments, detailed information of future art events can be displayed on the information platform in advance and the relevant art knowledge can be learned, so as to further promote the students' understanding of artworks, which could improve the cultivation effect of artistic literacy. This requires further experiments to confirm.

In addition, AI technology can also provide a solution for this [49]. Since students have different levels of artistic literacy, educational data mining (EDM) [50] can be used to digitize the viewing time, satisfaction, interest promotion, and other related information in the questionnaire, and machine learning technology can be used for feature analysis [51] and different methods and suggestions suitable for students with different characteristics to further understand artistic knowledge can be given so as to realize a personalized learning scheme. When the number of participating students is large enough and the feature sample is rich enough, ML can also be used for a certain degree of teaching intervention [52]. In the face of the COVID-19 pandemic, to maintain close student–teacher interactions, virtual classrooms have therefore become a reality [53]. Dr. Adjabi explored the effectiveness of combining both deep learning and traditional methods in addressing the SSFR issue and put forward a proposed new direction of the deep learning method based on semantic information [54], which may be verified in our following research in order to achieve a better learning effect.

Against the background of the global COVID-19 pandemic, due to the Chinese government's precise prevention and control strategy, most areas in China are at a low risk level, and local art venues are open as usual. Students can continue their art appreciation learning in their hometown through the off-campus learning mode investigated in this experiment. However, with multiple art appreciation projects to choose from at the same time, it is up to students to decide on the appropriate content. As pointed out by Broadbent and Schneider, SRL becomes particularly important when students are faced with concurrent complex achievement tasks and learning formats that require a high degree of autonomy [55,56], which require early theoretical courses to understand the characteristics of each art category; this helps inspire students' interest and provides knowledge support for them to make judgments.

The learning model investigated in this study is also applicable to other practical teaching courses with mixed and diverse attributes [57,58]. During sanitary confinement in Spain, Javier and others conducted virtual practical teaching for education students, which achieved the goal of helping to achieve teaching content and related competencies. The positioning technology applied in this study may be able to provide these future teachers some new inspiration [59]. Tang applied artificial intelligence technology to build a useful support system in the practical teaching of ideological and political theory courses [60]. The improvement goals of the theoretical courses he analyzed can be confirmed by the method of this research. Ann raised the concerns of teachers as producers of distance teaching content, and the use of social education resources in this study may be able to reduce the burden on teachers in practical teaching [61]. It also brings new challenges for school administrators and teachers [62,63]. As suggested by Jacques, teachers should not only improve their professional knowledge but also master more information-based teaching tools so that students can obtain the necessary knowledge [6].

## 6. Conclusions

In this experiment, with the support of ICT, students could make a relatively complete record of the process of participating in art appreciation activities. The recorded content included the venue of the art event, the name of the performance/exhibition, the level of the organizer, etc. Among them, the mobile positioning check-in accuracy rate reached 93.75%. Through the mobile communication positioning function and the check-in and check-out processes, the basic learning time can be determined. The satisfaction and review text in the check-out questionnaire can be used as a subjective reference for evaluating learning outcomes.

Through the questionnaire, it can be analyzed that satisfaction with art appreciation activities directly affects students' artistic interest, with a correlation coefficient of 0.78. Satisfaction was significantly correlated with psychological expectations (0.67) and art information obtained in the early stage (0.61). In view of the fact that the content of the questionnaire is related to the validity of the course credit, the content the students fill in will affect whether they get the credits. They will take the initiative to understand the background and content of the performance/exhibition before participating in art appreciation activities, and they will be more serious in the appreciation process. This promotes an in-depth understanding of the expressive meaning of works. Compared with art theory education carried out in classrooms, this can stimulate in-depth and enthusiastic active learning by students [64].

The off-campus art appreciation teaching mode based on ICT can meet the requirements of teaching evaluations and play a role in improving students' art appreciation levels and humanistic qualities. We believe that this is a feasible method for the sustainable development of aesthetic education in schools.

**Author Contributions:** Conceptualisation, B.S.; Methodology, B.S. and Z.W.; Visualisation, B.S., B.H. and R.L.; Validation, Z.W.; Formal analysis, B.H. and R.L.; Data curation, Z.W. and B.S.; Software, B.H. and R.L.; Supervision, J.Y. and R.Z.; writing—original draft preparation, B.S.; writing—review and editing, Y.C. All authors have read and agreed to the published version of the manuscript.

**Funding:** This research received no external funding.

**Institutional Review Board Statement:** Not applicable.

**Informed Consent Statement:** Not applicable.

**Data Availability Statement:** Not applicable.

**Conflicts of Interest:** The authors declare no conflict of interest.

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
