# Peer review of "Research on Open Practice Teaching of Off-Campus Art Appreciation Based on ICT"

_sustainability, doi:10.3390/su14074274_

Round 1

Reviewer 1 Report

The research topic can be resumed in its current form in Europe, in order to identify the position on arts education affected by the Covid pandemic 19. The results obtained can thus contribute to the reformulation of teaching strategies, from the perspective of the beneficiaries of the educational process.

Author Response

esteemed expert:

On behalf of all the contributing authors, I would like to express our sincere appreciations of your review comments concerning our manuscript. We have carefully revised the manuscript according to the othe reviewer’s comments。Including a further detailed description of the study introduction, a comparative discussion with other similar results, and some grammatical errors.

Thanks again for your professional and friendly advice.

Kind regards

Baoqing Song

Reviewer 2 Report

A summary outlining the aim of the paper,
"The authors propose that using ICT to carry out art appreciation education outside the school"

Review: 
Is the manuscript clear, relevant for the field, references adequate for this type of work, and structure is correct.

Author Response

(The authors gave the same response as above.)

Reviewer 3 Report

First of all, congratulations on the research carried out in the field of the arts and the promotion of heritage, both tangible (monuments, goods, etc.) and intangible (customs, music and folklore, etc.).

The article submitted for review, despite its brevity, is very well structured and responds perfectly to the skeleton of a research work: the introduction, which includes a section on educational policies, some results prior to the implementation of the research and the purpose of the research; the materials and methods, where the experimental procedure, the instruments used in the research and the sample are explained; the results, where the results of the experiment and the analysis of the data are collected; the discussion, where the results are compared with other research; and, finally, the conclusions, where an overall summary of the article is presented.

Nevertheless, I would like to make a series of recommendations that will give the article a greater scientific packaging and value with a view to its socialisation:

  • It would be interesting if the research objectives were synthetically explained using infinitive verbs and numbered (for example: SO1...; SO2...).
  • The discussion is rather brief. It would be interesting to explore other research and use more references to contrast and compare the results.
  • In the introductory part I would introduce a theoretical section to explain what you understand by art, the existing types of art and what aesthetic appreciation of art is. 

For all these reasons, and to reassess the work done, minor changes are recommended for publication. 

Author Response

esteemed expert:

Thank you for your nice comments on our article. According to your suggestions, we have supplemented several contents here and corrected several mistakes in our previous draft. The detailed point-by-point responses are listed below.Please see the attachment.

Thanks again for your professional and friendly advice.

Kind regards

Baoqing Song

Reviewer 4 Report

First of all, I would like to thank the authors for their contribution "Research on Open Practice Teaching of Off-Campus Art Appreciation Based on ICT".

I am generally very sympathetic towards the project of this paper.

Author(s) need to mention ethical issues for their study and the relations between science and society. I propose to add the following reference:

Petousi, V., & Sifaki, E. (2020). Contextualizing harm in the framework of research misconduct. Findings from discourse analysis of scientific publications, International Journal of Sustainable Development, 23(3/4), 149-174, DOI: 10.1504/IJSD.2020.10037655 https://www.inderscienceonline.com/doi/abs/10.1504/IJSD.2020.115206

The explanation of the research process and the results could be explained more thoroughly. 

I encourage you to more fully illuminate your analysis process.

Figure 3 Map of activity area distribution, is not very clear.

Please carefully proof-read spell check to eliminate grammatical errors.

I wish you the best of luck with the revisions of your manuscript.

Author Response

(The authors gave the same response as above.)
